# A Deep Learning Approach to Increase the Value of Satellite Data for PM$_{2.5}$ Monitoring in China

**Bo Li [1], Cheng Liu [2,3,4,5,*], Qihou Hu [3], Mingzhai Sun [2], Chengxin Zhang [2], Yizhi Zhu [2], Ting Liu [1], Yike Guo [6], Gregory R. Carmichael [7] and Meng Gao [8,9]**

1 School of Earth and Space Sciences, University of Science and Technology of China, Hefei 230026, China; bo520@mail.ustc.edu.cn (B.L.); tliu95@mail.ustc.edu.cn (T.L.)
2 Department of Precision Machinery and Precision Instrumentation, University of Science and Technology of China, Hefei 230027, China; mingzhai@ustc.edu.cn (M.S.); zcx2011@ustc.edu.cn (C.Z.); yzz2017@mail.ustc.edu.cn (Y.Z.)
3 Key Lab of Environmental Optics and Technology, Anhui Institute of Optics and Fine Mechanics, Hefei Institutes of Physical Science, Chinese Academy of Sciences, Hefei 230031, China; qhhu@aiofm.ac.cn
4 Center for Excellence in Regional Atmospheric Environment, Institute of Urban Environment, Chinese Academy of Sciences, Xiamen 361021, China
5 Key Laboratory of Precision Scientific Instrumentation of Anhui Higher Education Institutes, University of Science and Technology of China, Hefei 230027, China
6 Department of Computer Science, Hong Kong Baptist University, Hong Kong SAR, China; yikeguo@ust.hk
7 Department of Chemical and Biochemical Engineering, The University of Iowa, Iowa City, IA 52242, USA; gregory-carmichael@uiowa.edu
8 State Key Laboratory of Environmental and Biological Analysis, Department of Geography, Hong Kong Baptist University, Hong Kong SAR, China; mmgao2@hkbu.edu.hk
9 John A. Paulson School of Engineering and Applied Sciences, Harvard University, Cambridge, MA 02138, USA
* Correspondence: chliu81@ustc.edu.cn

**Abstract:** Limitations in the current capability of monitoring PM$_{2.5}$ adversely impact air quality management and health risk assessment of PM$_{2.5}$ exposure. Commonly, ground-based monitoring networks are established to measure the PM$_{2.5}$ concentrations in highly populated regions and protected areas such as national parks, yet large gaps exist in spatial coverage. Satellite-derived aerosol optical properties serve to complement the missing spatial information of ground-based monitoring networks. However, satellite remote sensing AODs are hampered under cloudy/hazy conditions or during nighttime. Here we strive to overcome the long-standing restriction that surface PM$_{2.5}$ cannot be obtained with satellite remote sensing under cloudy/hazy conditions or during nighttime. In this work, we introduce a deep spatiotemporal neural network (ST-NN) and demonstrate that it can artfully fill these observational gaps. We quantified the quantitative impact of input variables on the results using sensitivity and visual analysis of the model. This technique provides ground-level PM$_{2.5}$ concentrations with a high spatial resolution (0.01°) and 24-h temporal coverage, hour-by-hour, complete coverage. In central and eastern China, the 10-fold cross-validation results show that R$^2$ is between 0.8 and 0.9, and RMSE is between 6 and 26 (µg m$^{-3}$). The relative error varies in different concentration ranges and is generally less than 20%. Better constrained spatiotemporal distributions of PM$_{2.5}$ concentrations will contribute to improving health effects studies, atmospheric emission estimates, and air quality predictions.

**Keywords:** PM$_{2.5}$; air pollution; neural network; full-coverage

## 1. Introduction

The negative impact of ambient particles on humans and nature has become a global concern [1]. Aerosols play a role in shaping weather and climate systems [2] through their ability to absorb and scatter radiation, as well as their influence on cloud dynamics [3,4],

and particles with a diameter less than 2.5 μm ($PM_{2.5}$) are small enough to enter deeply into human lungs, posing the greatest short-term and long-term risks to health [5]. $PM_{2.5}$ can be created either from direct emissions or indirect sources resulting from complicated chemical reactions [6,7], which makes its composition complex [8]. This is also the reason why multiple countries are increasingly raising their attention to the regulation of $PM_{2.5}$ emissions and identification of its potential sources [9,10]. In addition, it can stay in the boundary layer for a few days and in the free troposphere for a few weeks [11–13]. It exits the atmosphere through precipitation, deposition, and other chemical reactions [14]. Simultaneously, under the influences of atmospheric diffusion and transmission, the diurnal variation of $PM_{2.5}$ can range from several μg m$^{-3}$ to many hundreds μg m$^{-3}$ within several hours, and appreciable differences in $PM_{2.5}$ concentrations can occur within several kilometers spatially [15–17].

Therefore, an accurate description of the spatiotemporal distribution characteristics of $PM_{2.5}$ is crucial. The spatiotemporal distribution of $PM_{2.5}$ is commonly obtained from ground sampling instruments or inferred from satellite remote sensing. Over the past several years, China has made remarkable progress in monitoring air quality, with the number of surface monitoring sites exceeding 1600 across the country in 2020 [18]. These sites are mainly concentrated in urban regions, while rural and rural-urban fringes, home to half of China's population, still lack systematic observation. Although the density of monitoring sites within urban areas is higher than that in rural areas (Table S1), some essential sources, especially point ones, can still be overlooked. Satellite aerosol optical properties are applied to complement the missing spatial information of monitoring networks. For example, the Himawari-8 launched by the Japan Meteorological Agency provides aerosol optical depth (AOD) at a 5-km spatial resolution every 10 min. Although satellite data is collected, only indirect indications of ground-level $PM_{2.5}$ concentrations are available. Aerosol optical depth (AOD) reflects the vertical distribution of aerosol extinction from the Earth's surface to the top of the atmosphere. In contrast, ground-based instruments solely detect $PM_{2.5}$ concentrations in proximity to the ground. When the long-range transmission of particles occurs above the ground, particulates do not gather around the ground level, and high AOD does not always coincide with high $PM_{2.5}$ concentrations [19]. Due to different characteristics of aerosol hygroscopic growth at different heights, satellite AOD and ground $PM_{2.5}$ vary due to their differences in seasonal attributes. For instance, in the summer of North China Plain, aerosols between 500 and 1000 m display a clear peak, resulting in differences between satellite observations of aerosol optical thickness and ground-based monitoring of $PM_{2.5}$ concentrations [20]. Furthermore, satellite observations are limited to cloud-free and haze-free scenes.

Currently, researchers make numerous efforts to derive ground-level $PM_{2.5}$ concentrations with satellite AOD and some observation results. The Gridpoint Statistical Interpolation (GSI) Three-Dimensional Variational (3DVAR) data assimilation system assimilates atmospheric variables from diversified approaches, including satellites, radars, and weather balloons, into the model. By solving a system of equations that minimize the cost function, the 3DVAR method is applied to adjust the initial conditions of the model (which may contain errors) to match observational results. Based on the derivation, the optimal adjustment of the initial parameters can be clarified. This system is usually adopted to link the changes in AOD to aerosol chemical compositions [21,22]. This approach is computationally expensive, and the performance can be degraded by the uncertainties in the operator itself [23]. Attempts to statistically infer ground-level $PM_{2.5}$ concentration from satellite AOD are also proposed [24,25]. Although spatiotemporal gaps were filled with predictions from chemical transport models or with AOD observed by multiple satellite sensors [24,25], predictions were obtained at relatively low temporal resolution (daily/monthly) [24–35]. Furthermore, errors in this approach may arise due to uncertainties in chemical transport modeling or unfavorable weather conditions such as cloudiness or haze. In addition, errors may also stem from uncertainties in the data or model system [24,25]. The method of monitoring AOD by geostationary orbit satellites relies on passive remote sensing using solar radiation,

and can only monitor AOD during daylight hours [36–38]. Improved predictions [39,40] suggested using a day-night band sensor (DNB), yet hourly variations remained unclear. A few studies addressed this issue by including temporal predictors, which could indicate the diurnal pattern of $PM_{2.5}$ [41,42]. However, the horizontal resolution of most of the input variables was seriously lower than the prediction, and these algorithms exhibited biases stemming from the limited data coverage of AOD retrievals under cloud cover, ice-covered surfaces, or during nighttime. Heavy haze can also be incorrectly classified as clouds in AOD retrievals [43]. For example, our statistical analyses suggest that the annual spatial coverage of satellite AOD is only 33% in North China, and even less in other concerned regions in China (Table S2). Better methods are thus needed to overcome these limitations, particularly for areas often impacted by thick clouds and severe haze pollution [44–46]. In this study, we construct a deep spatiotemporal neural network (ST-NN) model to derive ground-level $PM_{2.5}$ concentrations with inputs of satellite AOD, meteorological variables, and geographical information.

This work is structured as follows: In Section 2, existing materials and methods are introduced. After that, Section 3 describes the results with the ST-NN model. It obtains a high spatial and temporal resolution (0.01° and hourly) surface $PM_{2.5}$ even during nighttime and under cloudy or hazy conditions; it has conducted a rigorous verification and extensive discussion of the results.

## 2. Materials and Methods

We built a deep learning model to improve estimates of ground-level $PM_{2.5}$ concentrations, particularly for regions without sampling sites, and for conditions (cloudy, hazy, nighttime, etc.) where satellite retrievals are not available. Our research area is concentrated in the Middle East of China (Figure S3), with a time span of four years from 2017 to 2020. In Section 2.1, we introduced the input data of the model, in Sections 2.2 and 2.3, we introduced the model structure and training verification methods, and in Section 2.4, we introduced a sensitivity analysis method to quantitatively analyze the impact of input data on the results, and to visualize the relationship between input variables and target output variables.

### 2.1. Model Configuration Datasets

2.1.1. Datasets Description

The Chinese National Environmental Monitoring Center (CNMEC) network (http://www.cnemc.cn/, accessed on 1 January 2023) provided us with hourly ground-level $PM_{2.5}$ observations, which we used as the label and validation data. The measurements were obtained using the β X-ray method and the vibration balance method.

For the input datasets, the daily MODerate Resolution Imaging Spectroradiometer (MODIS) 3 km aerosol products [47] (https://doi.org/10.5067/MODIS/MOD04_3K.061, accessed on 1 January 2023) and the hourly 0.05° × 0.05° Himawari-8 AOD products are used [48] (https://doi.org/10.2151/jmsj.2018-039, accessed on 1 January 2023).

The original MODIS products were mapped onto regional grids of 0.05° × 0.05° resolution to ensure that the input data have the same grid center. Considering the diurnal variation of the solar zenith angles, only daytime satellite data (defined as 00:00–09:00 UTC) were used in this study. The reason for using two different types of satellite aerosol optical thickness is that the Himawari-8 satellite aerosol is effective in capturing the daily variation of aerosols [49], while the MODIS aerosol product has an aerosol optical thickness in different bands to capture information on the properties of aerosols and more accurate numerical results [47].

We use some geographic information data to explore its potential impact on $PM_{2.5}$. These inputs to the neural network include land cover types (MODIS land cover product at 0.05° × 0.05° resolution, yearly), the normalized difference vegetation index (MODIS, 0.05° × 0.05°, monthly), the enhanced vegetation index (MODIS, 0.05° × 0.05°, monthly), road network (originally meter level, www.openstreetmap.org; Last access: 10 July 2020),

point of interest data (POI), elevation data (1 km × 1 km, the Resource and Environment Science Data Center, RESDC, http://www.resdc.cn; Last access: 1 January 2021), and population/gross domestic product data from RESDC. These datasets, including road network, points of interest (POI), elevation, etc., were resampled to a uniform grid resolution of 0.01°. The resulting datasets were georeferenced to ensure they share a common grid center.

Meteorological variables are important factors affecting atmospheric aerosols [50] and meteorological conditions to heavy pollution episodes within the Beijing-Tianjin-Hebei region were larger than 50% from 2013 to 2017 [51]. Meteorological conditions accounted for 48% of $PM_{2.5}$ variations in Eastern China from 1998 to 2016 [52]. The weather fields (0.05° × 0.05°, hourly) simulated by the Weather Research and Forecasting (WRF) model version 4.0 with three nested domains [53,54] (https://www.mmm.ucar.edu/weather-research-and-forecasting-model; Last access: 15 May 2020). The initial conditions and boundary conditions of the meteorological fields were derived from the National Centers for Environment Prediction's (NCEP) 6-h final operational global (FNL) data with a spatial resolution of 0.25° × 0.25°.

The boundary field parameters were set at 26 mandatory levels from 1000 millibars to 10 millibars. These parameters include surface pressure, sea level pressure, geopotential height, temperature, sea surface temperature, soil values, ice cover, relative humidity, u- and v-winds, vertical motion, vorticity, and ozone. The parameters simulated by boundary field and WRF we used are shown in Tables 1 and S3. In order to evaluate the validation results of the WRF model and ground station, validation was carried out. Figure S1 shows the verification results of the main meteorological variables (monitoring of coincidence between ground meteorological stations and demand variables). The R of temperature and pressure is above 0.95, and the slope is close to 1. The correlation coefficient R between the wind speed simulated by the model and verified by ground meteorological stations is 0.65. The main reason for the difference between the two is that wind speed has strong local variability. The simulated value represents the average wind speed within a grid, while the observed value represents the result at a specific location. Descriptions and features of these considered datasets are listed in Tables 1 and S4–S6.

**Table 1.** Descriptions of considered variables.

| Product | Unit | Variable Definition | Spatial Resolution | Temporal Resolution |
|---|---|---|---|---|
| AOD | | Aerosol optical depth | 0.05° × 0.05 | 1 h |
| Tempc | °C | Temperature | 0.05° × 0.05° × 12 L | 1 h |
| RH | % | Relative Humidity | 0.05° × 0.05° × 12 L | 1 h |
| HPBL | m | Planetary Boundary Layer Height | 0.05° × 0.05° | 1 h |
| P | Hpa | Pressure | 0.05° × 0.05° × 12 L | 1 h |
| U | m/s | Wind Speed (U) | 0.05° × 0.05° × 12 L | 1 h |
| V | m/s | Wind Speed (V) | 0.05° × 0.05° × 12 L | 1 h |
| DEM | m | Digital Elevation Model | 0.01° × 0.01° | Annual |
| POI | | Point of Interest | 0.01° × 0.01° | Annual |
| Traffic Network | | Traffic Network | 0.01° × 0.01° | Annual |
| GDP | ¥/km² | Gross Domestic Product | 0.01° × 0.01° | Annual |
| TPOP | people/km² | population density | 0.01° × 0.01° | Annual |
| Land Cover Type | | Land Cover Type | 0.05° × 0.05° | Annual |
| EVI | | Enhanced Vegetation Index | 0.05° × 0.05° | Monthly |
| NDVI | | Normalized Difference Vegetation Index | 0.05° × 0.05° | Monthly |

### 2.1.2. Datasets Selection

The abundance of real-world data sources presents a challenge. To address this challenge, it is necessary to leverage prior research efforts and employ effective data filtering techniques to select appropriate input data. Such measures serve to mitigate the complexity of modeling and enable the achievement of desired outcomes within acceptable

computational cost constraints. First, we conduct data preselection and select data by checking the correlation between the input data variables and our target output (ground $PM_{2.5}$ concentration). All input features were used to examine the potential relationship with $PM_{2.5}$.

We employed the Maximal Information Coefficient (MIC) [55] to examine the correlation between variables (Table S7), which can detect both linear and nonlinear relationships among variables, but it involves high computational complexity. We also tested the Pearson correlation coefficient (Table S7), which is an effective tool for assessing the linear correlation between variables and has been widely used in this type of research [32,56,57]. If continuous variables have linear correlation, they are correlated, and linear correlation is a necessary but not sufficient condition for correlation. We use contingency tables to calculate the chi-square statistic to test the correlation and independence between discrete variables [58,59]. For variables such as roads, POI, etc., we consider their values to be constant over time within the time scale of the study. First, we use the k-mean method to cluster them for reducing dimensions. We classify the annual average data of CNEMC $PM_{2.5}$ according to WHO Global Air Quality Guidelines [60]; and estimate the independence and correlation of variables by calculating the chi-square statistic between variables. Only those parameters that pass the significance test ($\alpha < 0.05$) were selected (Table S7).

### 2.1.3. Datasets Filter

For CNEMC $PM_{2.5}$ data, we remove null data. Secondly, we also tested the performance of the model under different data filtering rules (Table S10).

All hourly data at a specific monitoring site were transformed into z scores, and then the transformed data ($Zi$) were removed if they met one of the following conditions: (1) the absolute $Zi$ was larger than 4 ($|Zi| > 4$), (2) the increment of $Zi$ from the previous hourly value was larger than 9 ($|Zi - Zi - 1| > 9$), or (3) the ratio of the z score to its centered moving average of order 3 (MA3) was larger than 2 ($3Zi/(Zi - 1 + Zi + Zi + 1) > 2$) [61]. For missing satellite AOD data, we use null values.

### 2.2. Data Preprocessing and ST-NN Model Configuration

Figure S2 displays the architecture of the deep learning ST-NN model. The input data includes satellite AOD, and meteorological and geographic information data (Tables 1 and S3). The central grid of all input data is the CNEMC site label location.

For the data of UTC 4:00~09:00, we use the previous four hours Himawari-8 AOD, the past days Himawari-8 AOD (daytime AOD of the past two days), and the past week Himawari-8 AOD (daily average) to formulate the influencing factors. For other times, we use the past days Himawari-8 AOD (daytime AOD of the past two days) and the past week Himawari-8 AOD (daily average) to formulate the influencing factors. MODIS AOD values retrieved at three bands were used for the past week's results. First, feature extraction (nonlinear transformation) is performed on each data layer, and then data fusion is performed. The data fusion process can be expressed as:

$$Z_{fusion} = \sum_{i=1}^{c} K_i \otimes X_{AOD_{Himawari-8_{current}}} + \sum_{i=1}^{c} K_i \otimes X_{AOD_{Himawari-8_{closeness}}} \sum_{i=1}^{c} K_i \otimes X_{AOD_{Himawari-8_{period}}} + \sum_{i=1}^{c} K_i \otimes X_{AOD_{MODIS}}$$

where $\otimes$ is convolution and $K_i$ is a learnable parameter.

Meteorological data were arranged as time series of the bottom model level and the vertical features at the current moment $t_0$ to include the influences of temporal and spatial evolution. For meteorological data and geographic information data, we use the same method to extract the data features and then perform data fusion. For data with different spatial resolutions, we use the upper sampling layer in the feature extraction process to make them have the same spatial size before data fusion.

We mainly use the Inception-Resnet block and pooling layers for feature extraction, which has been proven to be able to quickly and effectively mine the potential features of

multidimensional data (more details in Figure S2) [62]. PM$_{2.5}$ concentrations were then obtained by optimizing the Log-Cosh loss function below.

$$Loss = \frac{1}{n}\sum_{i=1}^{n} \log(\cosh(y_i^{predict} - y_i^{true})) \tag{1}$$

where $y_i^{predict}$ means the model predicted value and the $y_i^{true}$ represents the observations.

For small differences, the log-cosh loss performances are similar as $\frac{\left(y_i^{predict} - y_i^{true}\right)^2}{2}$, and for huge differences (at the beginning of model training), it is closer to abs $\left(y_i^{predict} - y_i^{true}\right) - \log(2)$.

We initialized all the layers with the built-in Keras glorot uniform initializer as 0, and the biases were initialized with 0. Due to the symmetry of the data, tanh was used as the activation function. We trained the networks for 64 epochs with a batch size of 4, and SGD (Stochastic Gradient Descent) optimizer with an exponential decay of the learning rate $\alpha$ as:

$$\alpha = \begin{cases} 0.001 & epoch <= 32 \\ 0.001 \times \exp(0.1 \times (32 - epoch)) & epoch > 32 \end{cases} \tag{2}$$

### 2.3. Training and Testing

Based on the number of samples, dimension of time, and dimension of space, the entire dataset (one year) was randomly sorted into ten sections, with nine sections for training and the rest for testing [63].

A 10-fold cross-validation (10-CV) has been proven to be a reasonable means to evaluate results, such as Table References cited in Table S9. We have a large dataset to avoid overfitting. Compared with other studies, our hourly full coverage ground PM$_{2.5}$ concentration prediction has greatly increased the amount of data. The sample size exceeds 1 million in each study area and year (Tables S11 and S12). Furthermore, we compared the results under different training and test proportions and showed that the 10-CV was reasonable in this study (Table S16). The shape of the input dataset is shown in Figure S2 and Table S3. For sample-based cross-validation, we randomly grouped all the data; for spatial-based cross-validation, we randomly grouped the data by site location; and for temporal-based cross-validation, we randomly grouped the data by time. The testing data does not participate in the model training process. The proposed model was implemented in Python 3.7 (https://www.python.org/, accessed on 1 January 2023) with a neural network library named Keras (https://keras.io/guides/, accessed on 1 January 2023) and TensorFlow as the backend (https://www.tensorflow.org/, accessed on 1 January 2023).

### 2.4. Sensitivity Analysis

The neural network model is generally considered a black box model, and we open the black box model through sensitivity analysis and visualization to analyze the quantitative impact of each input variable on the ground PM$_{2.5}$ concentration [64].

For M input variables {$X_a$: a $\in$ (1, ..., M)}, each input variable $X_a$ was divided into L levels, and $X_{a_j}$ denotes the jth level of $X_a$. For continuous variables, the L level is evenly divided into 10 parts according to the value range of input variables, and for classified variables, it is equal to the number of channels. N samples from the testing dataset were then selected randomly to replace $X_a$ values with $X_{a_j}$, and the mean responses of PM$_{2.5}$ ($\hat{y}_{a_j}$) were documented. With the spatial feature considered, the sensitivity of PM$_{2.5}$ to a continuous variable factor (AOD, meteorological variables, etc.) was examined by varying the factor $X_a$ through its range with $L$ levels but keeping the spatial pattern fixed. The $X_{a_j}$ was given as:

$$X_{a_j} = X_a - mean(X_a) + L_j \tag{3}$$

For classified factors, such as land use type and traffic networks, sensitivity analysis was conducted in the manner of unified feature type:

$$X_{a_j}[N, m, n, j] = \sum_{j=0}^{channels} X_a[N, m, n, j] \tag{4}$$

where $m$ and $n$ denote the location in spatial coordinates, while $j$ represents the location in category dimension.

Four metrics were calculated to evaluate the relative importance of input variables, namely range ($S_r$), gradient ($S_g$), variance ($S_v$), and Average Absolute Deviation (AAD) ($S_{AAD}$) [64]. For model inputs $X_a$, evaluation metrics were calculated with equations below:

$$S_r = \max(\widehat{y}_{a_j} : j \in \{1, \ldots, L\}) - \min(\widehat{y}_{a_j} : j \in \{1, \ldots, L\}) \tag{5}$$

$$S_g = \sum_{j=2}^{L} \left| \widehat{y}_{a_j} - \widehat{y}_{a_{j-1}} \right| / (L-1) \tag{6}$$

$$S_v = \sum_{j=1}^{L} \left( \widehat{y}_{a_j} - \overline{y}_{a_j} \right)^2 / (L-1) \tag{7}$$

$$S_{AAD} = \sum_{j=1}^{L} \left| \widehat{y}_{a_j} - \widetilde{y}_{a_j} \right| / (L-1) \tag{8}$$

where $\overline{y}_a$ and $\widehat{y}_a$ denote the mean and median of the responses. The relative importance ($r_a$) can be described as:

$$r_a = \varsigma_a / \sum_{i=1}^{M} \varsigma_i \tag{9}$$

where $\varsigma_a$ means the sensitivity measure for $X_a$ (e.g., range). In this study, the relative importance ($r_a$) was defined as a vector $\overrightarrow{r} = (r_1, r_2, \ldots, r_M)$.

The influence of errors in input data on predictions of PM$_{2.5}$ concentrations was explored with the equation below:

$$input\_data_A[l, i, j, m] = input\_data[l, i, j, m] \times (1 + relative\_error) \tag{10}$$

where $l, i, j, m$ represents the dimensions of input data ($l$: the batch size, $i$: latitude, $j$: longitude, $m$: channels), and relative error means the uniform distribution of upper and lower bounds of error.

### 3. Results

*3.1. ST-NN Model Reconstructs Observed Spatiotemporal (Both Daytime and Nighttime) Features of PM$_{2.5}$*

Our ST-NN model operated on three major data types, namely AOD, geographical factors, and spatiotemporal distributions of meteorological conditions (details are documented in the Section 2). It was built to improve the predictions of ground-level PM$_{2.5}$, particularly for regions without sampling sites, and for conditions (cloudy, hazy, nighttime, etc.) when satellite retrievals are not available. In this study, we focused on the most populated and concerned regions, North China. We also demonstrated that the proposed method can be easily applied to other parts of China including East China, South China, the Sichuan Basin, and the heavily polluted Shaanxi province (regions marked in Figure S3 and Table S8). The performance of the ST-NN model was cross-validated with respect to sampling selection, temporal variability, and spatial distribution. As displayed in Figure 1, our ST-NN model accurately captured the observed spatiotemporal variability of daytime PM$_{2.5}$, with regression slopes close to 1 and intercepts close to 0; the R$^2$ is above 0.8.

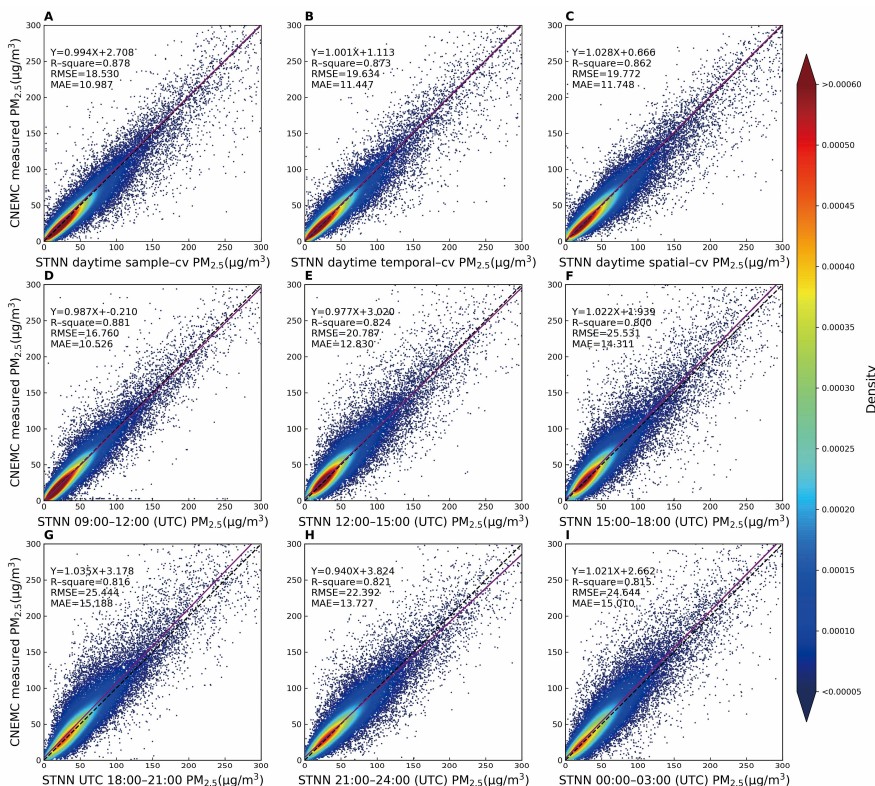

**Figure 1.** Density scatterplots of cross-validation with respect to sampling selection, temporal variability, and spatial distribution. (**A**) daytime, sample-based 10 cross-validation; (**B**) daytime, temporal-based 10 cross-validation; and (**C**) daytime, spatial-based 10 cross-validation; (**D–I**): spatial-based 10 cross-validation at different diurnal time slots (both daytime and nighttime). The fitting line is in purple, and the 1:1 standard line is the black dotted line.

Spatial variations of AOD at the past four hours, $t_{-3} \sim t_0$, were used as near-moment predictors for daytime $PM_{2.5}$. Daytime AOD values in the past two days and the past week were also used to formulate influencing factors across time. Multiple validation metrics, including $R^2$, root mean square error (RMSE, µg m$^{-3}$), and mean absolute error (MAE, µg m$^{-3}$), were calculated and listed in Figure 1 and Tables 1–4. $R^2$ values with respect to sampling selection, temporal variability, and spatial distribution in North China were generally above 0.85, RMSE was less than 26 µg m$^{-3}$ and MAE was less than 16 µg m$^{-3}$ (Figure 1), indicating the applicability of the model under various meteorological and geographical conditions. Its applications to other years and over other regions in China achieved similar good performance, and $R^2$ value reached even 0.90 when it is applied to Shaanxi Province for the year 2019 (Table 2). The root mean square error (RMSE) values were below 23 µg m$^{-3}$, as reported in Table 3. The mean absolute error (MAE) values were also below 16 µg m$^{-3}$, as shown in Table 4. Additionally, the slope values were found to be close to 1, as demonstrated in Table 5. We tested the relative uncertainty of the model (Figure S16). We find that the model validates worse in scenarios with lower and higher surface $PM_{2.5}$ concentrations, mainly due to the large observational uncertainty at low $PM_{2.5}$ concentrations and there is a small amount of data under high concentration conditions (Table S17).

Considering the potential spatial-temporal relationship between various variables and ground $PM_{2.5}$ under different conditions [9,11] and the lifetime of aerosol [10] (several days), daytime observed variations of AOD were used in the prediction of nighttime $PM_{2.5}$ (details documented in the Section 2). The capability in predicting the diurnal features of $PM_{2.5}$ was demonstrated in Figure 1D–I. Similar values of validation metrics were found for different time windows. $R^2$ values were generally above 0.80, and RMSE values were below 26 µg m$^{-3}$ for North China. Similar performances were found for other regions

also, and the performance of the model in predicting nighttime PM$_{2.5}$ did not exhibit a significant degradation from daytime (Table 2). Despite that satellite AOD retrievals are not available during nighttime, our ST-NN model provides a reasonably reliable prediction of PM$_{2.5}$ during nighttime. This is mainly attributed to the advantage of ST-NN in learning the dynamic transport and dissipation of particles under complex influences of meteorology, terrain, etc., which was exemplified by a haze episode occurring in North China in 2017 (Figure 2). From Figure S4, we can see that the characteristics of the PM$_{2.5}$ distribution in Beijing, PM$_{2.5}$ concentrations are influenced by topography and southwest transmission. The data are influenced by meteorological and aerosol data at $0.05°$. However, it can still be differentiated on a scale of $0.01°$.

**Table 2.** R-square values of cross-validation of the model with respect to spatial distribution.

| | 2017 | | 2018 | | 2019 | | 2020 | |
|---|---|---|---|---|---|---|---|---|
| | **Day** | **Night** | **Day** | **Night** | **Day** | **Night** | **Day** | **Night** |
| North China | 0.86 | 0.83 | 0.82 | 0.84 | 0.87 | 0.85 | 0.84 | 0.88 |
| East China | 0.81 | 0.82 | 0.86 | 0.85 | 0.83 | 0.86 | 0.86 | 0.85 |
| South China | 0.83 | 0.84 | 0.82 | 0.83 | 0.83 | 0.85 | 0.82 | 0.80 |
| Sichuan Basin | 0.84 | 0.85 | 0.82 | 0.80 | 0.89 | 0.89 | 0.87 | 0.83 |
| Shaanxi Province | 0.85 | 0.84 | 0.89 | 0.81 | 0.90 | 0.87 | 0.88 | 0.88 |

**Table 3.** RMSE of cross-validation with respect to spatial distribution.

| | 2017 | | 2018 | | 2019 | | 2020 | |
|---|---|---|---|---|---|---|---|---|
| | **Day** | **Night** | **Day** | **Night** | **Day** | **Night** | **Day** | **Night** |
| North China | 19.77 | 22.59 | 19.92 | 19.86 | 16.53 | 18.44 | 16.46 | 13.99 |
| East China | 16.15 | 16.51 | 13.09 | 14.04 | 13.19 | 12.13 | 9.88 | 9.47 |
| South China | 11.11 | 12.81 | 10.38 | 11.38 | 9.52 | 11.41 | 6.00 | 8.96 |
| Sichuan Basin | 14.80 | 17.52 | 13.90 | 18.51 | 10.28 | 11.86 | 8.03 | 10.74 |
| Shaanxi Province | 20.15 | 22.79 | 15.47 | 18.88 | 15.14 | 17.13 | 12.01 | 12.33 |

**Table 4.** MAE values of cross-validation of the model with respect to spatial distribution.

| | 2017 | | 2018 | | 2019 | | 2020 | |
|---|---|---|---|---|---|---|---|---|
| | **Day** | **Night** | **Day** | **Night** | **Day** | **Night** | **Day** | **Night** |
| North China | 11.75 | 15.02 | 11.91 | 12.15 | 8.94 | 10.86 | 8.93 | 8.24 |
| East China | 9.54 | 10.41 | 8.68 | 9.27 | 8.76 | 8.17 | 6.94 | 6.51 |
| South China | 6.82 | 7.84 | 6.77 | 8.22 | 6.53 | 7.34 | 4.14 | 6.01 |
| Sichuan Basin | 9.27 | 10.32 | 9.41 | 11.37 | 6.69 | 8.04 | 5.74 | 7.47 |
| Shaanxi Province | 12.48 | 13.96 | 9.98 | 11.97 | 9.15 | 10.48 | 7.80 | 8.03 |

**Table 5.** Slop values of cross-validation of the model with respect to spatial distribution.

| | 2017 | | 2018 | | 2019 | | 2020 | |
|---|---|---|---|---|---|---|---|---|
| | **Day** | **Night** | **Day** | **Night** | **Day** | **Night** | **Day** | **Night** |
| North China | 1.03 | 0.97 | 0.99 | 0.99 | 0.99 | 0.98 | 0.99 | 1.04 |
| East China | 1.07 | 1.00 | 0.97 | 1.05 | 0.96 | 1.00 | 0.98 | 1.04 |
| South China | 1.02 | 1.06 | 1.03 | 1.00 | 1.00 | 1.01 | 0.98 | 0.96 |
| Sichuan Basin | 1.01 | 1.00 | 1.01 | 1.03 | 1.03 | 1.04 | 1.04 | 1.02 |
| Shaanxi Province | 1.02 | 1.03 | 1.01 | 1.00 | 0.98 | 0.99 | 1.02 | 0.98 |

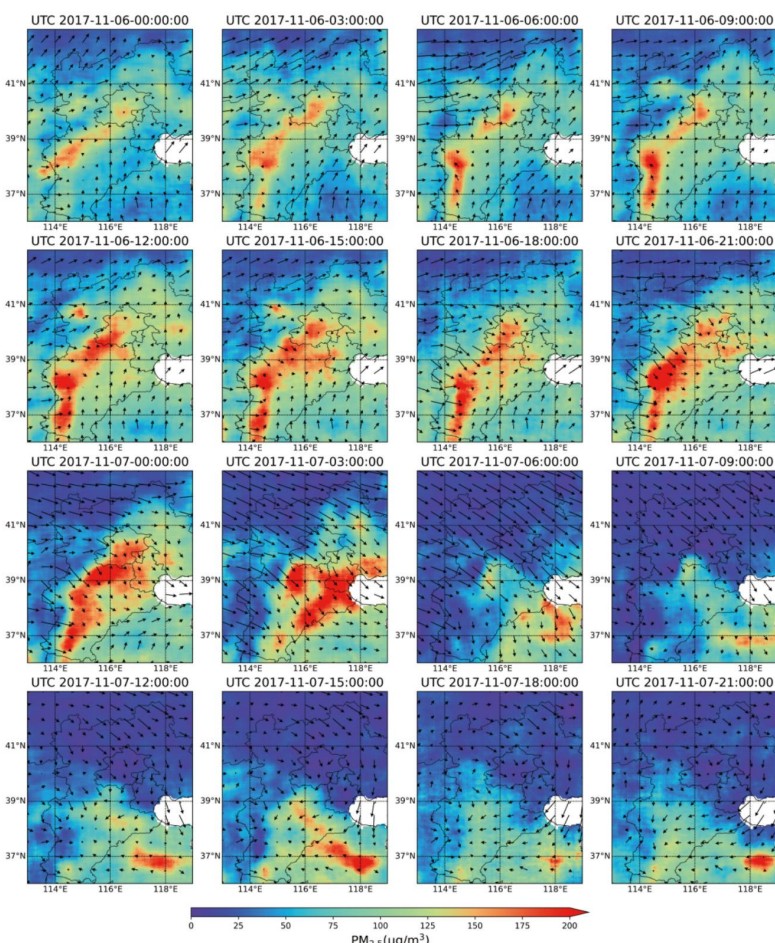

**Figure 2.** ST-NN model simulated haze episode on 16–17 November 2017. The spatial distribution of simulated near-surface PM$_{2.5}$ concentrations and wind fields.

In addition to cross-validation, independent validation of this ST-NN model was conducted with PM$_{2.5}$ concentrations observed at sites that were not included in the model training. The variability of the PM$_{2.5}$ concentrations at these independent stations were also accurately captured by our model, with R$^2$ values greater than 0.8, RMSE less than 24 $\mu$g m$^{-3}$, and MAE less than 15 $\mu$g m$^{-3}$ (Figure S5). Independent validation was conducted also with respect to the diurnal variation of PM$_{2.5}$. As indicated in Figure S6, the diurnal pattern of PM$_{2.5}$ over multiple independent stations across China was reproduced by the ST-NN model. The R$^2$ values greater than 0.8, RMSE less than 24 $\mu$g m$^{-3}$, MAE less than 14 $\mu$g m$^{-3}$ (Figure S6).

*3.2. Temporal and Spatial Block Cross-Validation*

To better assess the generalization of our model, additional spatial block cross-validation tests were carried out [63]. The East China region was selected for mask testing, and sites within the designated area were used as the validation dataset (see Figure S7). Our results show that as the mask area increases, the model's performance progressively worsens (see Table S13), but it still captures pollution events (see Figure S8). Even in North China, the most polluted region of China, the model produces good results under different weather conditions (see Figure S9). Furthermore, we evaluated the model's ability to capture PM$_{2.5}$ pollution in spatial block cross-validation using accuracy and precision metrics. Our results indicate that the accuracy rates were greater than 80%, and more than 75% of the sites had precision greater than 60% (see Table S15). This is mainly due to the increased spatial heterogeneity between the training and testing sites, resulting in a gradual reduction of characteristic data that can represent the target region in the training set. Figure S10 and

Table S14 show the validation results of time extrapolation, from which it can be seen that the model has a certain level of generalization on the timeline.

### 3.3. ST-NN Model Improves Prediction of $PM_{2.5}$ below Clouds and during Severe Haze

A prominent advantage of our ST-NN model is its competence in improving the prediction of $PM_{2.5}$ below clouds and during severe haze. Figure 3 displays satellite images during various episodes in different seasons when the North China region was obscured by clouds. In cloudy conditions, satellites fail to monitor ground-level aerosol pollution, while our ST-NN can fill these observation gaps and provide a complete distribution of $PM_{2.5}$ under cloudy conditions. Compared against ground-level observations, satisfactory performance was found (Figure 3), with $R^2$ values exceeding 0.82 in most cases. $PM_{2.5}$ hot spots in South Hebei and Shanxi as observed by the ground-level network were also reproduced by ST-NN. Figure S11 shows the overall relative error of cross-validation of the model under different cloud coverage.

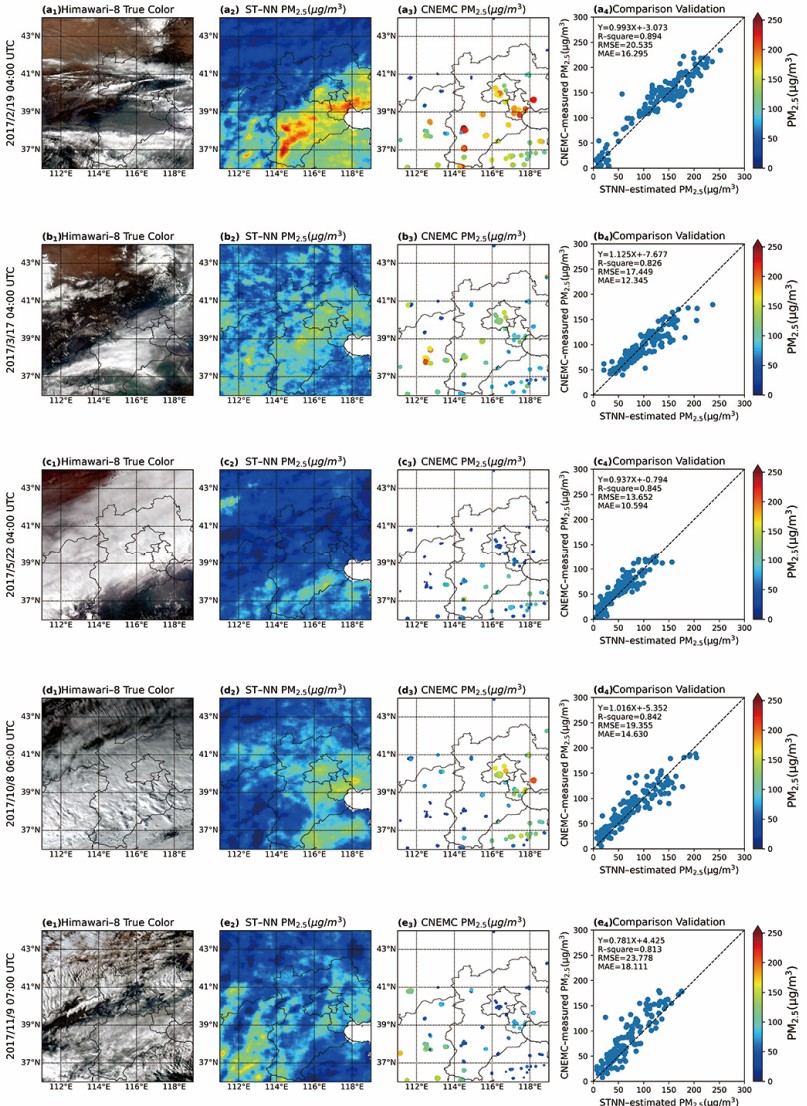

**Figure 3.** Performance of the ST-NN model at several cloudy moments (clouded data randomly selected from the results of time-based cross-validation). Left to right columns display true color images from Himawari-8 (five moments, (**a1**): 2017/2/19 04:00 UTC; (**b1**): 2017/3/17 04:00 UTC; (**c1**): 2017/5/22 04:00 UTC; (**d1**): 2017/10/8 06:00 UTC; (**e1**): 2017/11/9 07:00 UTC), ST-NN model predicted $PM_{2.5}$, at corresponding moments (**a2–e2**), CNEMC observed $PM_{2.5}$ at corresponding moments (**a3–e3**), and the scattered validation (**a4–e4**).

We further explored how cloudy conditions would influence the prediction of PM$_{2.5}$ concentrations. Figure 4 illustrates the predicted PM$_{2.5}$ with full coverage and with cloudy conditions removed for four metropolitan clusters in China. The MODIS Collection 6.1 Cloud mask products were used to track the cloudy conditions in this ST-NN model. Over the study period, 60% of the data in North China were affected by clouds. Heavy haze in China can also be misclassified as a cloud by the retrieval algorithm [43]. As a result, the influences of clouds on the prediction of PM$_{2.5}$ differ greatly across regions and seasons (Figure 4). Meteorological conditions have varying impacts on the concentration of PM$_{2.5}$ (fine particulate matter) across different regions (Figures S12 and S15). In North China and the Sichuan Basin region, mean PM$_{2.5}$ concentrations with cloudy periods removed exhibit lower values than the full coverage annual mean (Figure 4c,o). On the contrary, negative differences were identified for South China (Figure 4k), suggesting different driving factors for these regions. In cloudy scenes, PM$_{2.5}$ concentrations exhibited lower values when relative humidity (RH) > 60% in South China (Figure S12c). This could be related to cloud-precipitation-related wet removal of air pollutants. Conversely, PM$_{2.5}$ concentrations in North China were biased low using only cloud-free scenes in North China, as indicated with lower satellite-observed AOD in cloud-free scenes (Figure 5a–d). Such underestimation mainly occurred during wintertime (Figure 5).

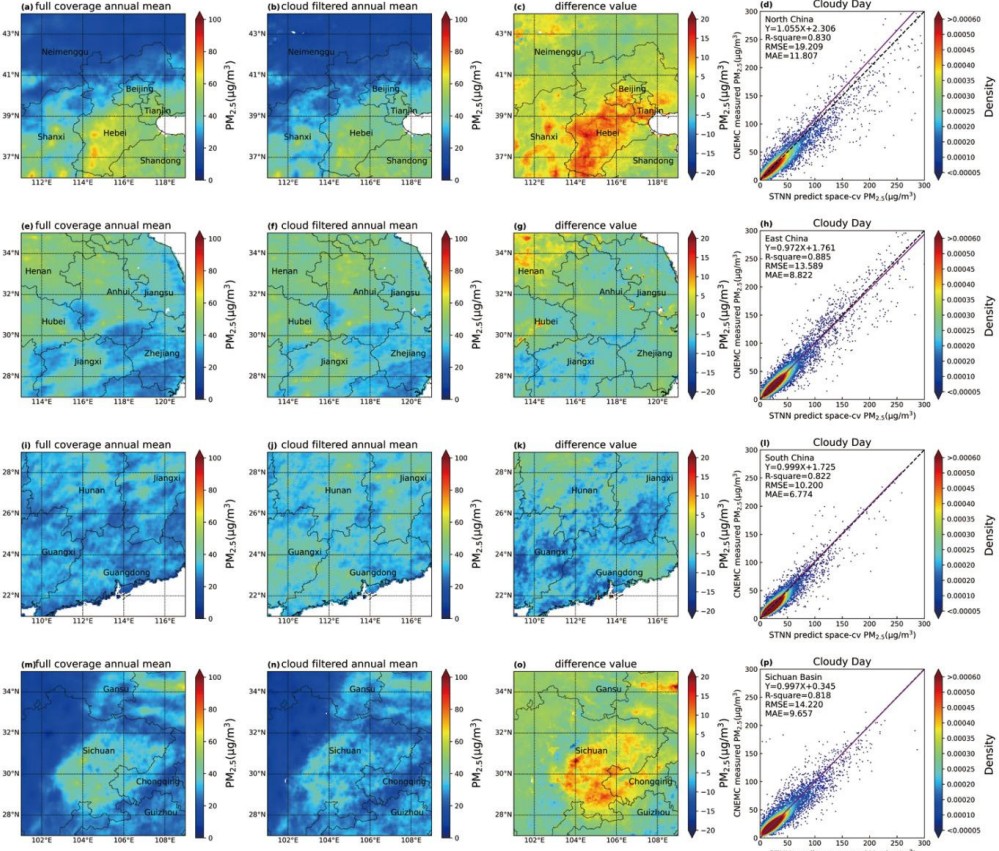

**Figure 4.** ST-NN model predicted annual mean PM$_{2.5}$ concentrations in 2017 and validation under cloudy conditions in 2017. ST-NN model predicted full coverage annual mean (**a,e,i,m**) for North China, East China, South China, and Sichuan Basin, respectively); predicted annual mean with MODIS marked cloudy conditions removed (**b,f,j,n**); the differences between predictions with full coverage and those with MODIS marked cloudy conditions removed (**c,g,k,o**); Cross-validation with respect to spatial distribution under conditions at stations that were not considered in training. (**d,h,l,p**) show the validation results under different cloud regions.

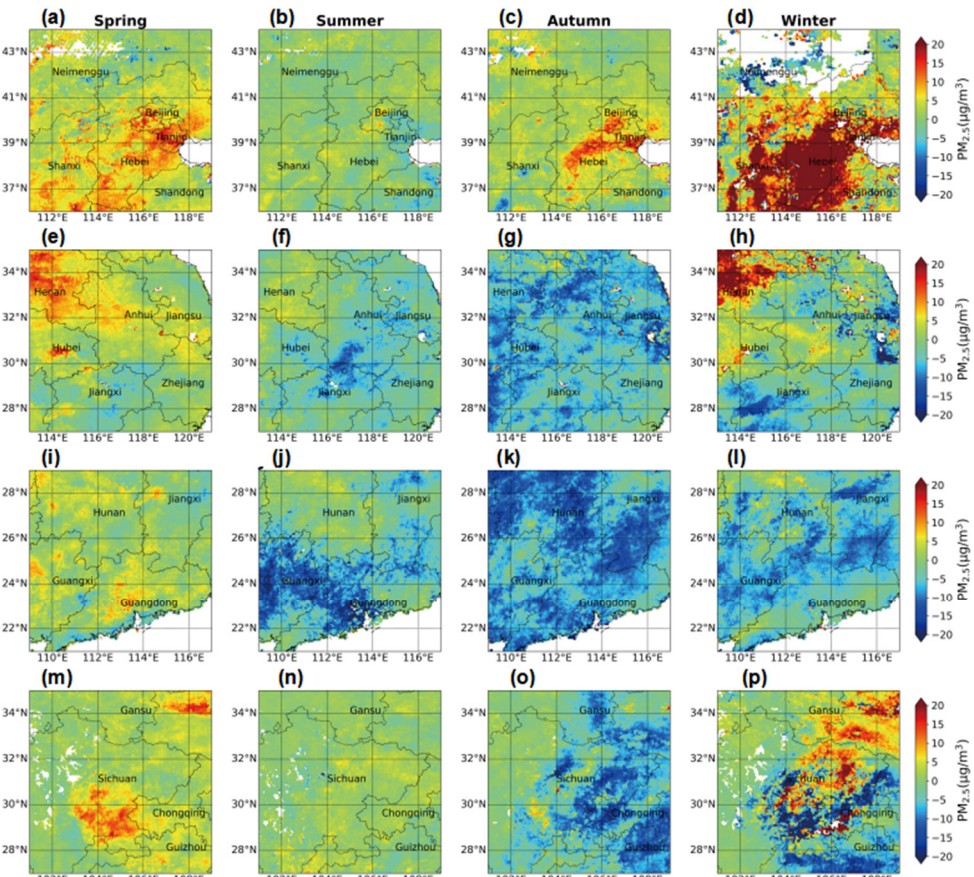

**Figure 5.** Seasonal distribution of the difference in PM$_{2.5}$ concentrations between full coverage data and cloud mask filtered data for four metropolitan regions. (**a–d**) North China. (**e–h**) East China. (**i–l**) South China. (**m–p**) Sichuan Basin.

The CNEMC surface measurements further indicated that PM$_{2.5}$ concentrations were biased low in cloud-free scenes in North China, but biased high in South China, consistent with our ST-NN predictions (Table 6). Over the entire study period, 83% of the data in winter in North China were marked as cloudy, higher than those in other seasons (60%) (Table S2). This is mainly related to the occurrences of snow/ice or severe haze [43]. Figure 4d,h,l,p further justified that ground-level PM$_{2.5}$ under cloudy conditions could be well predicted by our ST-NN model in four metropolitan regions, with high correlation coefficients and low errors. Cross-validation also suggested that our ST-NN model can give valid results under clouds (Figure S11). Different regions are affected differently by cloud cover, with warmer and more humid regions such as Eastern and Southern China, where errors increase as cloud cover increases, while in dryer regions such as Northern China, the effect of cloud cover has little impact on the results, possibly indicating a potential relationship between surface PM$_{2.5}$ concentrations and cloud formation processes.

**Table 6.** CNMEC measured and ST-NN predicted PM$_{2.5}$ (µg m$^{-3}$) concentrations in 2017.

|  | CNEMC Annual Mean | CNEMC Cloud Filtered Mean | ST-NN Annual Mean | ST-NN Cloud Filtered Mean |
|---|---|---|---|---|
| North China | 58.30 | 43.57 | 33.84 | 29.58 |
| East China | 48.57 | 44.33 | 38.49 | 40.75 |
| South China | 38.27 | 46.14 | 29.77 | 35.94 |
| Sichuan Basin | 46.88 | 36.48 | 25.80 | 24.63 |
| Shaanxi Province | 51.15 | 40.47 | 33.54 | 30.23 |

*3.4. ST-NN Model Offers Better Regional Representation of PM$_{2.5}$*

As the CNEMC stations are concentrated in urban areas (Figure S13), using only CNEMC data to estimate regional PM$_{2.5}$ concentration would result in an overestimation. As indicated in Table 7, mean PM$_{2.5}$ concentrations at CNEMC stations agree better with the mean over densely populated areas, but are higher than the mean over sparsely populated areas and therefore also higher than the mean over the entire region. This further emphasizes that CNEMC observations might not be able to reflect the pollution in the suburbs and accurately show the overall pollution condition in a region.

**Table 7.** ST-NN model estimated PM$_{2.5}$ ($\mu g \, m^{-3}$) concentrations under different population densities.

| | North China | East China | South China | Sichuan Basin | Shaanxi Province |
|---|---|---|---|---|---|
| CNEMC | 58.65 | 48.65 | 38.71 | 46.11 | 54.08 |
| Populated Regions (>500 people/km$^2$) | 53.40 | 43.20 | 31.31 | 38.55 | 46.38 |
| Moderately populated (<500 people/km$^2$) | 29.43 | 36.36 | 27.72 | 24.36 | 32.04 |
| All areas | 34.12 | 38.53 | 28.11 | 25.80 | 33.52 |

## 4. Discussion

Several studies [24–35,37] have explored the prediction of ground-level PM$_{2.5}$ concentrations using statistical methods, as summarized in Table S9. These studies have utilized multiple data sources to extract potential features and estimate near-surface PM$_{2.5}$ concentrations. However, most of these studies have provided low temporal resolution results (daily) and lack full spatiotemporal coverage. One longstanding challenge in using satellite aerosol optical depth (AOD) data has been the inability to constrain surface PM$_{2.5}$ concentrations under cloudy conditions, during nighttime, or severe haze [21].

The previous research on full coverage of PM$_{2.5}$ achieved good results. Zhang [65] used XGboost to obtain $R^2 \sim 0.87$, Tang [42] used Two Stage RF to obtain $R^2 \sim 0.86$ validation results, and Xiao [25] used Two Stage RF to obtain $R^2 \sim 0.81$ validation results. The spatiotemporal full coverage PM$_{2.5}$ results obtained by the ST-NN model are superior to most previous studies summarized in Table S9. At the same time, it has a better level of generalization in both time and space.

In our previous work, we used an atmospheric physical and chemical model to simulate meteorological fields with higher spatiotemporal resolution ($0.05°$ and hourly) compared to previous studies ($0.125/0.25°$ and 3/6 h), and we applied spatiotemporal neural networks to fully extract the spatiotemporal features of the data and the spatiotemporal correlations of aerosol itself. As a result, we obtained 24-h full-coverage results. The main influencing factors for PM$_{2.5}$ in different regions of China vary, and meteorological factors play an important role in different regions. However, different regions have different sources of pollution, and their performance under the influence of meteorological conditions varies. In the north, hygroscopic growth is an important factor contributing to PM$_{2.5}$ pollution, while in the south, wet deposition reduces PM$_{2.5}$ concentration, and the impact of wind fields on PM$_{2.5}$ pollution is significantly affected by pollution sources and topography. The complex terrain increases the deviation of input meteorological data, and the uneven distribution of national monitoring stations results in spatial data heterogeneity (Table S1). This leads to differences in the validation accuracy of different stations. Furthermore, as the satellite aerosol optical depth has the highest weight, the quality of data varies in different regions (reference). These factors contribute to the production of result errors, and we have quantitatively analyzed their impact (Figure S17). Our ST-NN model relies on the regional transport features of air pollution, and thus it may be challenging to track very small point sources.

### 5. Conclusions

In this study, we fully used the spatiotemporal features of aerosol and simulated the dynamic evolution of aerosols under complex influences of meteorology, terrain, etc. Without night and hazy/cloud data has been overcome here with an advanced statistical method. The capability of the built ST-NN model in predicting $PM_{2.5}$ below clouds and during nighttime is mainly due to the consideration of the spatiotemporal variation influencing meteorological/geographical factors and the dynamic evolution of aerosols. We select the input data through the correlation test based on the existing data at the beginning. Of course, such as the real-time emissions of some factory vehicles, building construction, etc. will also affect the ground $PM_{2.5}$ concentration. Sampling selection, temporal variation, and spatial distribution-based cross-validation demonstrated that the method presented here is skilled in providing reliable ground-level $PM_{2.5}$ concentrations with high spatial resolution ($0.01°$) and 24-h temporal coverage, which is challenging especially for heavily polluted regions. Independent validations were also conducted for cloudy conditions and night time, and no degradation of performance was found. We examined the importance of satellite-observed AOD in the prediction of $PM_{2.5}$ during both daytime and nighttime using four sensitivity measures [64], namely range $S_r$, gradient $S_g$, variance $S_v$, and average absolute deviation from the median $S_{AAD}$. These four sensitivity parameters measure the impact of input variables on the results. $S_r$ is sensitive to changes in the data range, $S_g$ is sensitive to changes in data gradient, $S_v$ is sensitive to data variance, and $S_{AAD}$ is more stable. We use sensitivity analysis and visualization to open the black box model of neural networks [64]. We evaluated the impact of each input parameter on the results. AOD accounts for more than 30% of the weight throughout the day, and the relative significance exhibits slightly higher values during nighttime (Table 8), emphasizing the importance of AOD observations in nighttime predictions. Land cover type and meteorological variables also play important roles in the dynamic evolution of $PM_{2.5}$ in North China and other regions, as illustrated in Figure S14. The effects of the key variables on surface $PM_{2.5}$ concentrations are given in Figure S15. However, the model tends to better capture moderately polluted conditions, as the relative errors exhibit relatively larger values when observed $PM_{2.5}$ concentrations are above 350 µg m$^{-3}$ or below 20 µg m$^{-3}$ (Figure S16). The relatively poor capability of our ST-NN model in capturing these extremely low or high values is mainly attributed to the rarity of these conditions and the small sampling size (North China: 0.34‰, East China: 0.059‰, South China: 0.068‰, Sichuan Basin: 0.048‰, Shaanxi Province: 0.25‰). This is also the limitation of the data model. The lack of sample size in a specific case leads to a certain deviation in the extreme case estimation of the model space to the real-world space. Similar uncertainties of the model might be raised by the errors in model input data. Random errors were added to the input data to explore how they would influence the errors of predicted $PM_{2.5}$. Similarly, the quality of AOD data was essential within a very broad range of uncertainty (Figure S17). When errors of other inputs grow (>20%), the accuracy of prediction would also be significantly degraded (Figure S17). We also examined how the input data quality control process would affect the accuracy, and a negligible role was found (Tables S10 and S11). During the development of ST-NN models for different regions in China, the loss function decreased similarly, while the decreasing speed and convergence values varied among regions due to differences in the size and feature of data (Figure S18). We noticed that the performance of the model varied across regions and seasons, which might be also related to the distinct spatiotemporal features of $PM_{2.5}$ (Figure S19), and the associated meteorological/geographical characteristics in different regions. The uneven distribution of CNEMC sites might also play a role (Figure S13).

**Table 8.** The importance of AOD in the prediction of $PM_{2.5}$, as indicated with sensitivity measures ($R_r$, $R_g$, $R_v$, and $R_{AAD}$).

|  | 00:00–06:00 (UTC) | 06:00–12:00 (UTC) | 12:00–18:00 (UTC) | 18:00–24:00 (UTC) | Day | Night |
|---|---|---|---|---|---|---|
| $R_r$ | 0.34 | 0.31 | 0.33 | 0.32 | 0.34 | 0.36 |
| $R_g$ | 0.42 | 0.39 | 0.41 | 0.40 | 0.42 | 0.44 |
| $R_v$ | 0.36 | 0.27 | 0.32 | 0.30 | 0.35 | 0.39 |
| $R_{AAD}$ | 0.36 | 0.31 | 0.33 | 0.32 | 0.35 | 0.37 |

In the future, we expect to obtain more accurate and higher resolution meteorological and satellite remote-sensing data. Additionally, with further exploration of the relationship between clouds and aerosols [22], we also anticipate effectively correlating observed cloud information with aerosols to improve results. As the most concerning $PM_{2.5}$ pollution events continue to accumulate over time, the increasing sample size of observed pollution events will make simulation more accurate, and generate additional samples of pollution events through the use of generative adversarial networks. Currently, the selection of training and testing sets is based on random sampling, but the uneven distribution of national monitoring stations themselves results in greater bias in results in certain geographic conditions. In the future, we will select training and sample sets based on geographic features to make the training sets more representative.

**Supplementary Materials:** The following supporting information can be downloaded at: https://www.mdpi.com/article/10.3390/rs15153724/s1. References [24–42,56,57,65–68] are cited in the Supplementary Materials.

**Author Contributions:** C.L. and M.G. conceived the research, and M.S. provided technical support. B.L. conducted model simulations and analyzed results; M.G., G.R.C., Q.H., C.Z., Y.Z., T.L. and Y.G. assisted with the discussion; all authors contributed to the final interpretation and writing of the manuscript with major contributions by C.L., B.L. and M.G. All authors have read and agreed to the published version of the manuscript.

**Funding:** This study was supported by grants from the Strategic Priority Research Program of the Chinese Academy of Sciences (No. XDA23020301), the National Key Research and Development Program of China (2022YFC3710101), the National Natural Science Foundation of China (41977184), the Key Research Program of Frontier Sciences, CAS (No. ZDBS-LY-DQC008), the Youth Innovation Promotion Association of CAS (2021443), the HFIPS Director's Fund (BJPY2022B07 and YZJJQY202303).

**Data Availability Statement:** All data needed to evaluate the conclusions in the paper are present in the paper and the Supplementary Materials. Additional data related to this paper may be requested from the authors.

**Acknowledgments:** We acknowledge NASA for providing the MODIS measurements and Japan Meteorological Agency for offering Himawari-8 aerosol optical depth (AOD). We acknowledge Martin G. Schultz for his valuable suggestions on this work.

**Conflicts of Interest:** The authors declare no conflict of interest.

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
