# Peer review of "A Deep Learning Approach to Increase the Value of Satellite Data for PM2.5 Monitoring in China"

_remotesensing, doi:10.3390/rs15153724_

Round 1
Reviewer 1 Report
The study developed an AI technique for estimating PM2.5 in cloudy and haze and nighttime from meteorological variables and satellite derived AOD including the current and previous times. The inputs are washed and cleaned prior to the training and the AOD data from different satellites are also fused. The PM2.5 from training data are verified through 10-fold validation and shows a higher R-square. However, the study is lack of validation on the model when it is used for prediction at other times. These independent validations should be done to warrant the robustness of the AI technique. My detailed comments are as follows:
(1) About of half of abstract are spent on discerptions of general PM monitoring issues. Indeed, many of previous studies have been developed for retrieving PM in cloudy and nighttime conditions. I recommend authors to focus more on the issues related to previous techniques and what is new in the proposed algorithm.
(2) In general, the science paper like this should be written in more passive voices to focus on the actions being taken and the results are derived from the analysis. The pronoun like “we” are used in so many places.
(3) Introduction seems to be written with AI tool like chat-gpt. The sentence to sentence is seem to poorly connected though the grammars are correct. Also, the paragraph to paragraph is also lack of connection. Also, those statements with GSI are irrelevant to the main topics.
(4) Authors should define a flow chart of this AI system and how is its result compared with other studies such as Liu and Weng’s early results from Himawari and MODIS data.
(5) Authors should make an independent validation. The sample data from the same time period is reserved for validation does not mean much in my view.
Still have some typoes and grammar mistakes. Too many active voices and pronouns are used in writing
Reviewer 2 Report
In the paper, authors established a deep model of spatiotemporal neural network model to fill the observation gap of PM2.5. In general, influencing factors are comprehensively considered in the model, and the description of methods and results is clear. One suggestion to authors: please consider the applicable conditions of the model and carry out necessary discussion in the conclusion.
Author Response
Reply:
We appreciate the valuable suggestions provided by the reviewer. In our conclusion, we have extensively discussed the applicable conditions of our model. Figure S16 demonstrates that the relative error of the model significantly increases under extremely low (PM2.5 concentration less than 5) and extremely high (PM2.5 concentration greater than 350) pollution conditions. Moreover, as observed in Figures S7 and S10, the model's error gradually increases as the representativeness of labels decreases. These findings highlight the importance of exercising caution when utilizing the model for spatiotemporal extrapolation.
Reviewer 3 Report
Please find below some minor corrections and suggestions :
1. In section 2.1.3 (Datasets filter), the authors could provide some reasoning in the text for the choice of the thresholds for the z score values in the various conditions
2. In figure 1, it is not clear what is the meaning of subfigures (a),(b) and (c) . What does daytime sample-cv, daytime temporal-cv, daytime spatial-cv correspond to? This is not explained in the text either.
3. lines 497-500: It must be removed
4. line 523: what is the meaning of (reference)?
5. In table S3 , meteorology paramameters are given twice with different shapes
6. lines 543-544, what is the meaning of the parameters Sr,Sg,Sv and SAAD? Please provide some information in the text. The same applies to Table 7
7. Figure S10 and Figure S12 are not cited in the text
8. Table S14 is not cited in the manuscript
9. In the Discussion , it would be helpful if the authors provided a comparison between their results and other results in the literature (at least for daytime PM2.5) obtained with deep learning or with classical statistical approaches.
Round 2
Reviewer 1 Report
I am now satisfied with authors responses
N/A